# PRSet: Pathway-based polygenic risk score analyses and software

**Shing Wan Choi**[1⊕], **Judit García-González**[1⊕], **Yunfeng Ruan**[2], **Hei Man Wu**[1], **Christian Porras**[1], **Jessica Johnson**[1], **Bipolar Disorder Working group of the Psychiatric Genomics Consortium**[¶], **Clive J. Hoggart**[1], **Paul F. O'Reilly**[1]*

**1** Department of Genetics and Genomic Sciences, Icahn School of Medicine, Mount Sinai, New York City, New York, United States of America, **2** The Broad Institute of MIT and Harvard, Cambridge, Massachusetts, United States of America

⊕ These authors contributed equally to this work.
¶ Membership of the Bipolar Disorder Working group of the Psychiatric Genomics Consortium is provided in Supporting Information file S1 Acknowledgements.
* paul.oreilly@mssm.edu

**Data Availability Statement:** All relevant data are within the manuscript and its Supporting Information files. The scripts used to perform quality control on UK Biobank data are available at https://gitlab.com/choishingwan/ukb_process. The

## Abstract

Polygenic risk scores (PRSs) have been among the leading advances in biomedicine in recent years. As a proxy of genetic liability, PRSs are utilised across multiple fields and applications. While numerous statistical and machine learning methods have been developed to optimise their predictive accuracy, these typically distil genetic liability to a single number based on aggregation of an individual's genome-wide risk alleles. This results in a key loss of information about an individual's genetic profile, which could be critical given the functional sub-structure of the genome and the heterogeneity of complex disease. In this manuscript, we introduce a 'pathway polygenic' paradigm of disease risk, in which multiple genetic liabilities underlie complex diseases, rather than a single genome-wide liability. We describe a method and accompanying software, PRSet, for computing and analysing pathway-based PRSs, in which polygenic scores are calculated across genomic pathways for each individual. We evaluate the potential of pathway PRSs in two distinct ways, creating two major sections: (1) In the first section, we benchmark PRSet as a pathway enrichment tool, evaluating its capacity to capture GWAS signal in pathways. We find that for target sample sizes of >10,000 individuals, pathway PRSs have similar power for evaluating pathway enrichment as leading methods MAGMA and LD score regression, with the distinct advantage of providing *individual-level* estimates of genetic liability for each pathway -opening up a range of pathway-based PRS applications, (2) In the second section, we evaluate the performance of pathway PRSs for disease stratification. We show that using a supervised disease stratification approach, pathway PRSs (computed by PRSet) outperform two standard genome-wide PRSs (computed by C+T and lassosum) for classifying disease subtypes in 20 of 21 scenarios tested. As the definition and functional annotation of pathways becomes increasingly refined, we expect pathway PRSs to offer key insights into the heterogeneity of complex disease and treatment response, to generate biologically tractable therapeutic targets from polygenic signal, and, ultimately, to provide a powerful path to precision medicine.

scripts used in the current study are available at https://gitlab.com/choishingwan/prset_analyses and https://gitlab.com/JuditGG/bd_subtypes. PRSet is a module within PRSice and is available on github repository [https://github.com/choishingwan/PRSice].

**Funding:** Support includes grants from the UK Medical Research Council (MR/N015746/1) and the National Institute of Health (R01MH122866) to PFO, which covered salaries for PFO, SWC, YR, HMW, and JGG. This work was supported in part through the computational resources and staff expertise provided by Scientific Computing at the Icahn School of Medicine at Mount Sinai, specifically the Minerva Supercomputer and the Mount Sinai Data Ark data commons, which was supported by the Office of Research Infrastructure of the National Institutes of Health under award number S10OD026880. The funders had no role in study design, data collection and analysis, decision to publish, or preparation of the manuscript.

**Competing interests:** All authors disclose no competing interest.

## Author summary

As proxies of genetic liability, polygenic risk scores (PRSs) are being increasingly applied in multiple fields and designs. However, most leading methods to compute PRSs are based on aggregating genome-wide genotypes to a single number for each individual. While these genome-wide PRSs are demonstrably useful, aggregating risk according to the functional sub-structure of the genome may be more powerful for many PRS applications.

Here we introduce a new method and accompanying software, PRSet, to calculate and analyse pathway-based PRSs, in which polygenic scores are computed across different genomic pathways for each individual. We find that pathway-based PRSs have similar power for evaluating pathway enrichment as the leading methods designed for the task (e.g. MAGMA), while pathway PRSs offer the distinct advantage of providing individual-level estimates of genetic liability for each pathway. All applications of genome-wide PRSs are available to pathway-specific PRS, but we expect the latter to offer greater insights into the heterogeneity of complex disease. We therefore investigate the performance of pathway PRSs versus genome-wide PRS methods to stratify patients of heterogeneous diseases into more homogeneous sub-groups, as a proof-of-principle of their potential utility to provide more powerful paths to precision medicine.

## Introduction

As proxies for genetic liability to human traits or diseases [1], polygenic risk scores (PRSs) have been applied in numerous applications, including prediction of disease risk [2–7], patient stratification [8], investigation of treatment response [9–12] and genetically-informed experimental perturbation [13,14]. Most leading PRS methods, including those that incorporate functional annotation [15,16], are based on the classical polygenic model of disease, which assumes that individuals lie on a linear spectrum from low to high genetic risk and that summarises an individual's genetic profile to a single value estimate of liability [17]. While this model has proven sufficiently accurate for utility across a range of applications, it incurs substantial loss of information about an individual's genetic profile, such as how the burden of genetic risk varies across different biological processes and pathways. This information may be more informative for many applications of PRS, such as patient stratification and prediction of treatment response.

In this study, we introduce a new polygenic risk score approach that accounts for genomic sub-structure, constitutes an extension to the classic polygenic model of disease, and may better reflect disease heterogeneity (**Fig 1A**). Instead of aggregating the estimated effects of risk alleles across the entire genome, pathway-based PRSs aggregate risk alleles across *k* pathways (or gene sets) separately. Therefore, rather than a single genome-wide PRS, each individual has *k* PRSs corresponding to *k* pathways across the genome. Well-defined pathways should reflect the encoding of different biological functions, separable in the same way that different environmental risk factors, such as smoking or dietary factors, are considered separately in epidemiological prediction models. From this perspective, GWAS results can be considered a composite of signal corresponding to function encoded by different genomic pathways (**Fig 1B**).

We begin by introducing PRSet, a method and accompanying software for computing and analysing pathway-based PRSs, where pathways can be defined in multiple ways, including by existing databases (e.g. KEGG, REACTOME [19,20]), or by analytically derived modules of

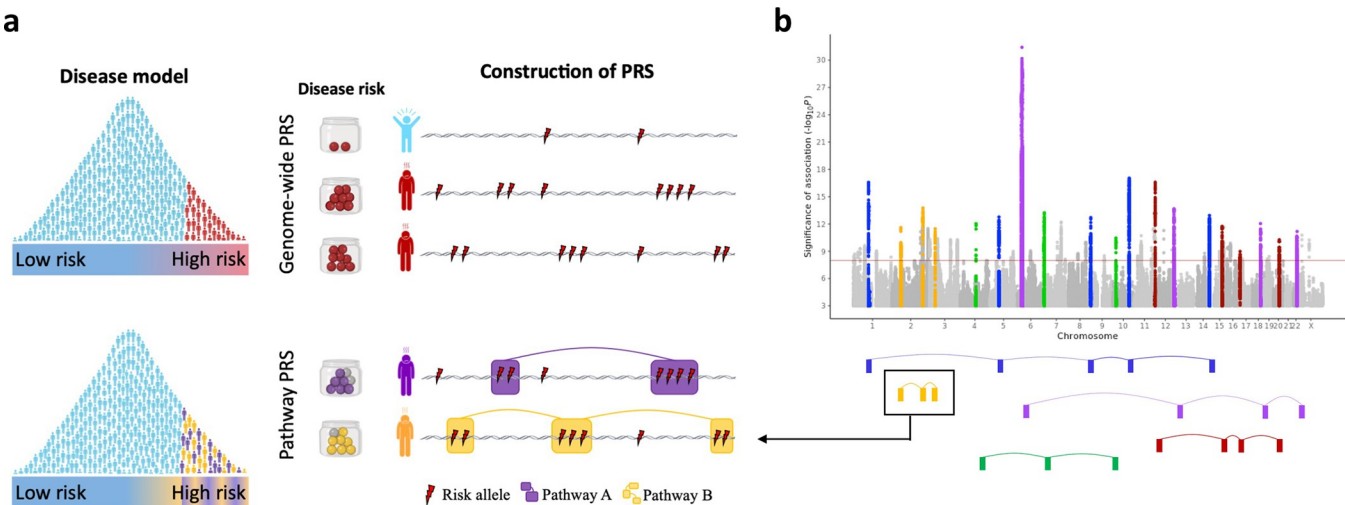

**Fig 1. The pathway polygenic risk score approach.** Coloured boxes represent genes, lines link genes that are within the same genomic pathway. **A,** Upper model: Classical polygenic model of disease, in which individuals lie on a linear spectrum from low to high risk and genome-wide PRSs are constructed as the sum of risk alleles across the genome. Disease risk depicted by the Jar model [18]. Lower model: Pathway polygenic model of disease, in which there are multiple liabilities and PRSs are constructed by aggregating risk alleles over different genomic pathways. **B,** GWAS results Manhattan plot illustrated as a hypothetical composite of signals, where each signal corresponds to an alternative functional route to disease. Pathways that only make a small contribution to disease risk across the population, or a contribution in a small fraction of individuals (e.g. nicotine receptor pathway in those individuals who smoke), are likely to harbour risk variants of relatively small effect. Figure partially created with BioRender.com.

e.g. gene co-expression, cell-type specific expression or protein-protein interactions, or from functional output of experimental perturbations [21–23].

Our results are separated into two main sections. In the first section, we assess how well PRSs capture GWAS risk signal across pathways, since a key concern in application of PRS computed over relatively short genomic regions is whether they are sufficiently powered to capture GWAS risk signal and, thus, be useful. Here we show, for the first time, that the performance of PRSs in capturing genetic signal at the pathway-level is comparable to that of leading pathway enrichment methods MAGMA [24] and LD score regression (LDSC) [25] when applied to target sample sizes of at least 10,000 individuals. Therefore, pathway PRSs may be powered for a range of other applications for which genome-wide PRSs are presently used. In the second section of the results, we test this premise using real data, performing a head-to-head performance comparison of pathway PRSs versus genome-wide PRSs for disease stratification into subtypes of inflammatory bowel disease, bipolar disorder, multiple major diseases according to their comorbidities, as well as stratification in to "pseudo subtypes" that correspond to diseases and their combinations (see **Results**). We show that pathway PRSs outperform standard genome-wide PRS alternatives, C+T (implemented in PRSice-2 [26]) and lassosum [27], for stratification into subtypes, often by a wide margin. We expect the power of pathway PRSs to improve substantially in the future with improved definition of pathways, more accurate functional annotation of genes, and with further development of pathway PRS methodology. Our new method and accompanying software, PRSet, builds on the popular PRSice genome-wide PRS tool [26,28] and is likewise user-friendly, fast, intuitive and openly available.

## Results

### PRSet model overview

Our PRSet method for calculating pathway-based PRSs leverages the classical genome-wide PRS method [1]—clumping + thresholding (C+T)—to calculate $k$ PRSs corresponding to $k$

genomic pathways for an individual $i$, as follows:

$$PRS_{ik} = \sum_{j=1}^{m_k} \beta_j G_{ij}$$

where $m_k$ is the number of clumped SNPs in pathway $k$, $\beta_j$ is the SNP effect size estimated from a GWAS on the studied phenotype, and $G_{ij}$ is the genotype of individual $i$ in pathway $j$, which comprises multiple genes across the genome defined, for example, according to biochemical knowledge [19,20] or gene co-expression networks [21,22].

In contrast to the genome-wide C+T method, where SNPs are clumped across the whole genome, PRSet performs clumping on each pathway independently, which retains pathway signal and account for correlation between SNPs in nearby genes of the same pathway. This also ensures that the SNPs present in multiple pathways are counted for each individual pathway. Since performing clumping on each pathway independently can be computationally intensive, PRSet utilizes a bit-flag system where the membership of a SNP in a pathway is represented as 1 if the SNP is in a pathway, or 0 if the SNP is outside of a pathway. During clumping, SNPs are removed from a pathway (the bit-flag of a SNP changes from 1 to 0) if and only if the SNPs are in the same pathway and the same clumping window as the index SNP (**S1 Fig**). This allows PRSet to perform the pathway clumping without repeating the entire clumping procedure.

Many applications of standard genome-wide PRSs can be adapted to pathway PRSs, the analyses of which can be evaluated and reported similarly. For example, each pathway PRS can be tested for association with a phenotype of interest in a target sample by regressing the phenotype on the PRS, as in standard PRS analyses. Additionally, PRSet can evaluate pathway enrichment by computing an empirical "competitive" $P$-value, which accounts for pathway size via the number of (clumped) SNPs included in the pathway using a permutation procedure (see **Methods**).

When calculating and analysing pathway PRSs, some extra considerations are needed: Firstly, the definition and annotation of pathways is critical for the interpretation of pathway PRS results. For this reason, PRSet gives the user great flexibility to input any list of SNPs or genes composing a pathway. For example, the user can extend the 3' and 5' gene boundaries to capture SNPs outside of genes, or can add distal SNPs with inferred regulatory effects on the genes. Secondly, the use of the $P$-value thresholding procedure is dependent on the use-case. For example, while $P$-value thresholding is not performed in pathway enrichment analyses, it is performed to optimize prediction in the disease subtyping application of this study (see **Methods**).

## Evaluating the power of PRSet using a pathway enrichment approach

In this section, we benchmark the power of pathway PRSs for assessing pathway enrichment, versus MAGMA and LDSC. It is important to note that (1) PRSet is not optimised as a pathway enrichment tool, but these analyses are performed to assess how well pathway PRSs capture GWAS signal and, thus, their potential for wider use, (2) Although the three methods assess the enrichment of GWAS signal across pathways, they use different statistical models and rely on different assumptions (**Methods** and **Fig 2A**). Since the ranking of pathways according to their GWAS signal enrichment is typically the outcome of most interest in enrichment analyses, we evaluate method performance using the Kendall's correlation between the rank of pathways based on their known enrichment and the rank according to the enrichment inferred by the methods. We use a range of comparisons that define pathways in different ways, and that can be separated into (i) those that use canonical pathways, and (ii) those that define pathways by tissue and cell-type specific gene expression.

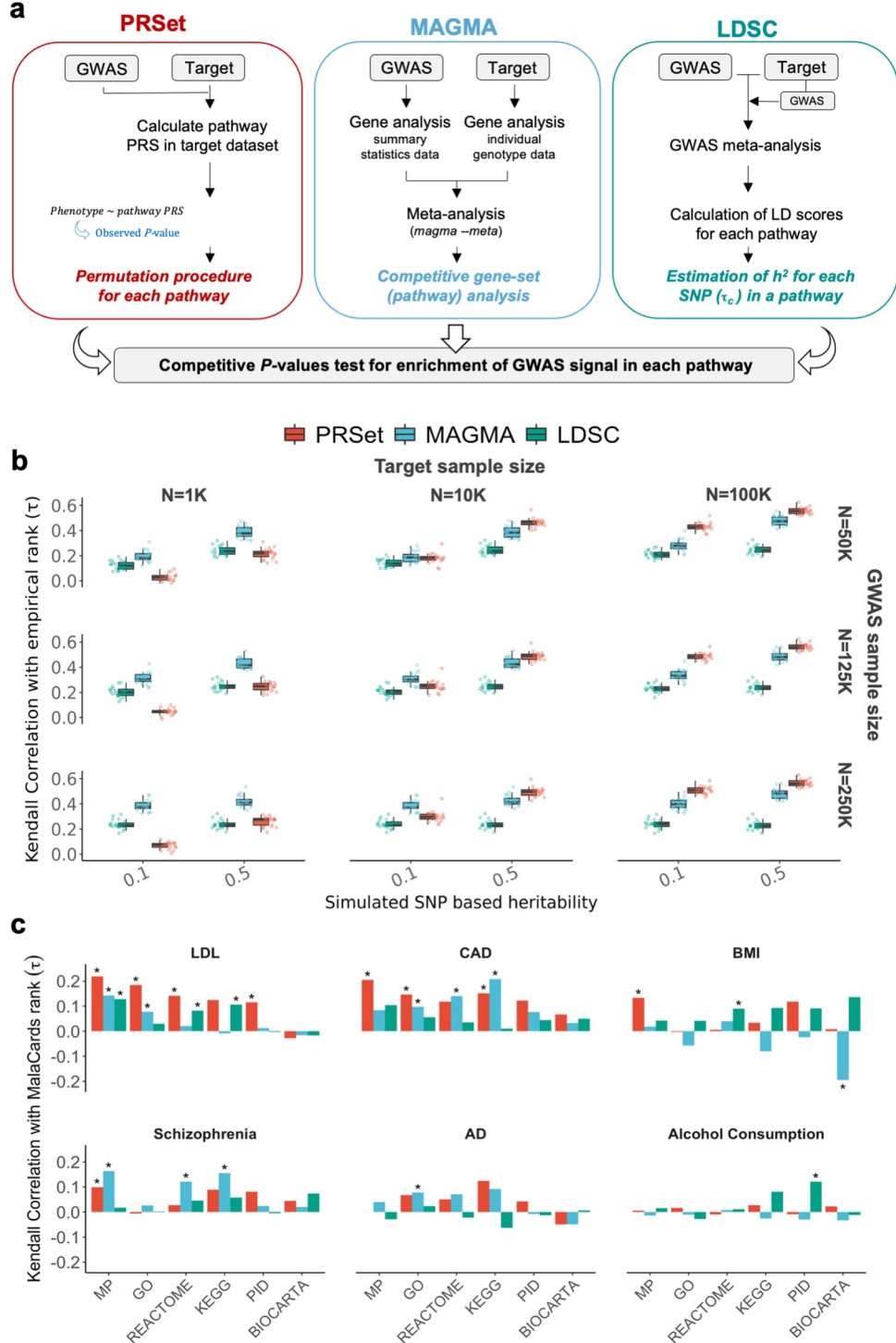

**Fig 2. Evaluating pathway enrichment using canonical pathways. A,** Schematic overview of PRSet, MAGMA and LDSC for assessing pathway enrichment**. B,** Pathway enrichment results for simulations with 50 random causal pathways. Performance is defined as the Kendall correlation between the pathway ranks based on competitive *P*-values of enrichment computed by each software and the empirical pathway ranks based on the true (simulated) effects across the pathways. Boxplots illustrate the values of Kendall rank correlation coefficients (τ) for PRSet, MAGMA and LDSC for each combination of heritability (h$^2$ = 0.1, 0.5), base sample size used in GWAS n = (50*k*, 125*k*, 250*k*), and target sample size n = (1K, 10K, 100K). **C,** Pathway enrichment results using real data for six diseases. Kendall correlation coefficients (τ) between pathway ranks based on competitive *P*-values of enrichment computed by each software and pathway ranks based on *MalaCards* disease relevance scores. *Empirical *P*-value < 0.05. MP, Mouse Genome

Database; GO, Gene Ontology database; KEGG, Kyoto Encyclopaedia of Genes and Genomes; PID, Pathway
Interaction Database. AD, Alzheimer's disease.

### Canonical pathways

In this sub-section, 4,079 pathways are defined using six publicly available databases (Biocarta [29], Pathway Interaction Database [30], Reactome [19], Mouse Genome Database [31], KEGG [20] and GO [32,33]) and pathway enrichment of genetic signal is tested by: (i) a simulation study, (ii) real data using *MalaCards* gene scores (**Methods**).

First, we simulated quantitative traits of different heritability ($h^2$ = 0.1, 0.5) using real genotype data of UK Biobank individuals, with a number of pathways (50 and also 4,050) randomly selected from the six pathway databases to contain between 1% and 30% (in step sizes of 1%) causal SNPs, with all other pathways containing no causal variants, ensuring pathways of varying enrichment of causal signal (**Methods**). GWASs were then performed on 50*k*, 125*k* and 250*k* individuals and their simulated traits, and an additional 1*k*, 10*k* and 100*k* individuals were selected as target data. A target data set is required for PRSet analyses (comprising individuals for which PRS are calculated), but not for MAGMA and LDSC. To ensure that the input data were identical for all methods, PRSet, MAGMA and LDSC were applied to both GWAS and target data sets to test for pathway enrichment. We ran MAGMA on GWAS summary statistics and target data separately, and meta-analysed the results. For LDSC, which takes summary statistic data as input only, we calculated a GWAS on the target data and meta-analysed the results with the base GWAS. The meta-analysis summary statistics were used as input for LDSC (**Fig 2A** and **Methods**). Subsequently, we ranked the pathways by their inferred enrichment and calculated the Kendall's correlations between the inferred and the known simulated enrichments to evaluate the methods' performance. This process was repeated 20 times.

**Fig 2B** and **Table A in S1 Tables** displays the results for simulations with 50 pathways, showing best overall performance for MAGMA (Median Kendall τ = 0.51), then PRSet (Median Kendall τ = 0.42) and then LDSC (Median Kendall τ = 0.38). All methods perform better with larger $h^2$, in particular MAGMA and PRSet. Whereas MAGMA and LDSC results remain similar across target sample sizes, PRSet performance increases with larger target sample sizes, being the best-performing method for the 100k target data. These differences in performance as a function of target sample size are likely due to differences in the impact that increasing sample size has on each of the different models: In the case of PRSet, the calculation of the competitive *P*-value is directly affected by the target sample size, since the nominal and null *P*-values are obtained from the regression model of *Phenotype ~ PRS*. Here the number of observations corresponds to the number of individuals in the target sample and directly impacts the estimation of *P*-values.

**S2A Fig** displays the results for simulations with 4,050 pathways, where the three methods show lower correlations with the known simulated enrichment. Under this scenario, the heritability tagged by each SNP is smaller (since $h^2$ is spread across 4,050 pathways instead of 50 pathways), therefore the correlation between the inferred and known signals is lower.

Next, we apply the three methods to the real data of UK Biobank, and that of publicly available GWASs, across six traits: low-density lipoproteins, coronary artery disease, schizophrenia, body mass index, Alzheimer's disease (proxy status) and alcohol consumption. Since the true GWAS signal enrichment of each pathway is unknown, we produce a *disease relevance score* for each pathway by summing *MalaCards* gene scores (**Methods**), which assign values to genes based on systematic phenotype-specific text-mining of the literature (note that most genes are assigned a *MalaCards* score of 0).

In **Fig 2C** and **Table B in S1 Tables**, we report the Kendall's correlations between the rank of the pathways according to the enrichment estimated by the three methods versus the *MalaCards* disease relevance scores. While the three methods show broadly similar results (**Fig 2C**), with PRSet having the highest median correlation ($\tau = 0.078$) between its pathway enrichment ranks and those of the *MalaCards* scores, followed by MAGMA ($\tau = 0.050$) and LDSC ($\tau = 0.043$), the performance varies widely depending on pathway resource (**Fig 2C**) and trait (**S2B Fig**). There are 24 significant results, 15 of them corresponding to low-density lipoproteins and coronary artery disease, 5 are obtained when using LDSC, 9 with PRSet and 10 with MAGMA. However, one of the MAGMA significant results (BMI calculated using BIO-CARTA) had a marginal P-value (0.012) and was in the unexpected direction ($\tau = -0.19$). We also repeated the analysis removing all genes with *MalaCards* scores greater than 0 to examine evidence of pathway enrichment among genes not yet highlighted in the literature and found that the correlations were eliminated (**S1 Text**). This may indicate that the methods have limited power to identify weak effects across pathways, or that only a modest fraction of genes in pathways influences disease contribution to risk.

## Pathways defined using tissue/cell-type specificity

To further interrogate the power of PRS to capture genetic signal at the pathway-level compared to MAGMA and LDSC, we compared the performance of the methods in tissue/cell-type expression specificity analyses using the approach introduced in *Skene et al* 2018 [34]. This approach tests whether genes that are specifically expressed in certain tissues or cells are enriched for GWAS signal–as evaluated by MAGMA and LDSC (and here PRSet)–and are thus implicated in disease aetiology. Following the approach of *Skene et al*, genes are grouped into 11 quantiles of increasing expression-specificity based on expression reported across 47 bulk-tissues and 24 brain cell-types (**Methods**). Next, we tested two models to evaluate the enrichment of GWAS signal in increasingly-specific tissue/cell-types. One model assesses the enrichment of the genes in the top quantile, which we refer to as the 'top quantile' test model, while the other assesses the linear trend of enrichment and is referred to as the 'linear' test model (**Methods**).

Here we perform these analyses in the same data and traits used in the previous section. In the absence of well-established roles for individual tissue/cell-types in these outcomes, we sought *a priori* candidates from two domain experts for each outcome to provide an agnostic way to evaluate the performance of the different methods in this setting (**Methods**).

We observed significant associations between expert opinion (**Table C in S1 Tables**) and the tissue-type specificity results (**Table D in S1 Tables**), although results varied substantially depending on the pathway method and test model used (**Fig 3A and Table E in S1 Tables**). The enrichment of GWAS signal across tissues was strongest for schizophrenia (**Fig 3B–3C and Fig A in S2 Text**) and body mass index (**Fig 3 and Fig B in S2 Text**), in which MAGMA and LDSC had a higher correlation with expert opinion than PRSet. However, in Alzheimer's disease (**Fig 3A and Fig C in S2 Text**) and coronary artery disease (**Fig 3A and Fig D in S2 Text**), PRSet enrichment results showed higher correlation with expert opinion than MAGMA and LDSC.

The associations relating to the cell-type specific analyses were relatively weak (**Fig 3D**), with significant correlation results between expert opinion and cell-type enrichment only observed for MAGMA and PRSet in relation to schizophrenia. For Alzheimer's disease, the strongest and only significant enrichment result was that of PRSet implicating microglia using the top quantile test model (**Fig 3E**), which is notable since microglia has been extensively linked to Alzheimer's disease aetiology in the literature [35]. However, individual results

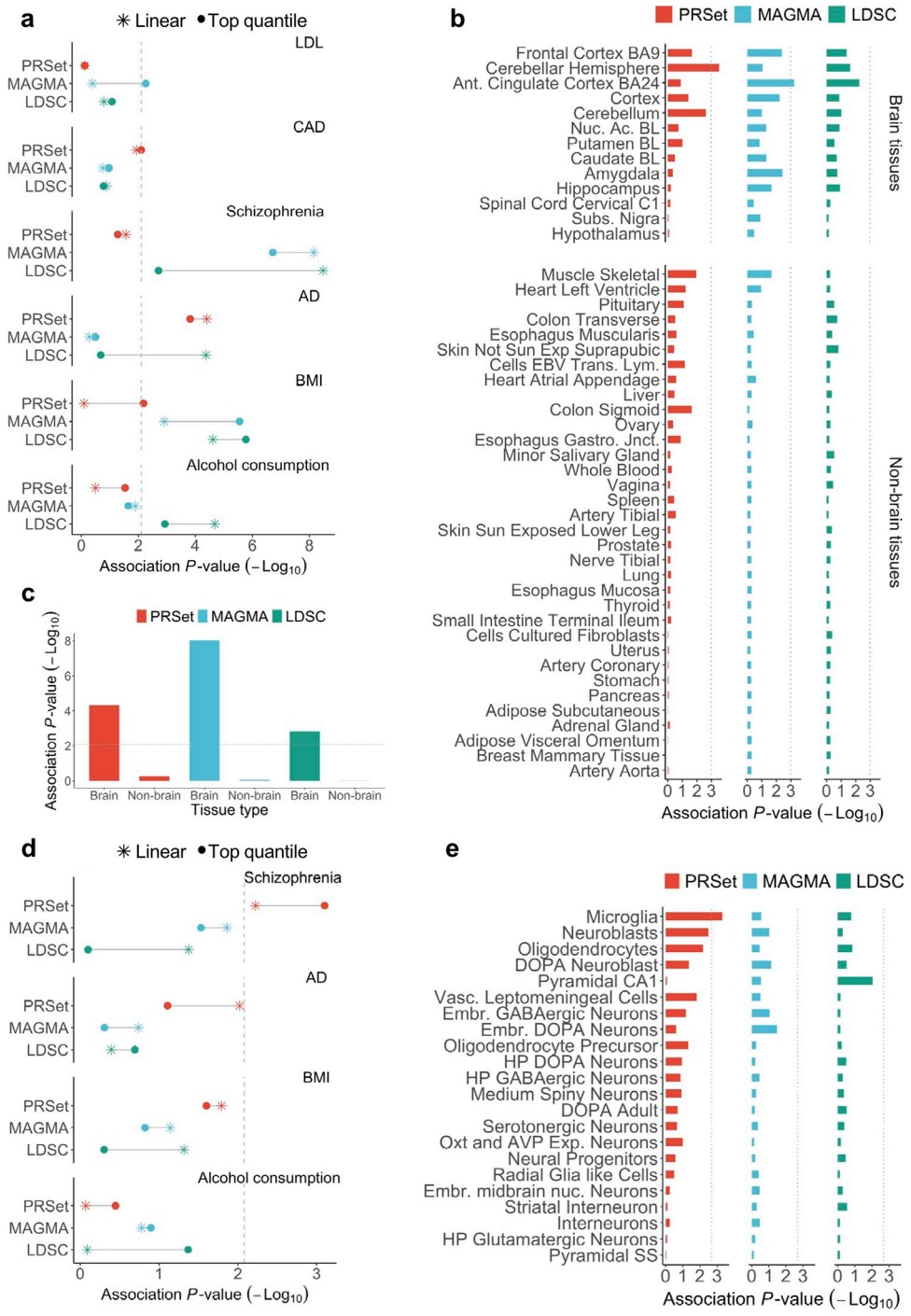

**Fig 3. Performance of PRSet, MAGMA and LDSC for ranking of pathways defined by tissue-type and cell-type expression specificity. A,** Association between pathway enrichment *P*-value and expert opinion of tissue relevance for each software and six diseases. Colours indicate the software used to calculate enrichment: Red, PRSet; Blue, MAGMA; Green, LDSC. Results are shown for both the *top quantile* and *linear* specificity test methods (**Methods**). Dashed line corresponds to Bonferroni significance threshold of 0.05 for 6 tests (3 methods x 2 test models). **B,** Pathway enrichment results for schizophrenia under the top quantile test model. Bar plots show enrichment *P-value* for each tissue and pathway method. Dashed line corresponds to Bonferroni significance threshold for 47 tissues (-log10(0.05/ 47) = 2.97). Ant, Anterior; Nuc. Ac., Nucleus Accumbens; BL, Basal Ganglia; Subs, Substantia; Exp, Exposed; EBV Trans. Lym., Epstein-Barr virus transformed lymphocytes; Gastro. Jnct, Gastroesophageal Junction. **C,** Enrichment of schizophrenia signal is higher in brain tissues vs non-brain tissues under the top quantile test model. Bar plots show the meta-analysis enrichment *P*-value using the Fisher's method for brain vs non-brain tissue and method. Dashed line

corresponds to Bonferroni significance threshold for the 6 tests conducted. **D,** Associations between pathway enrichment *P*-value and expert opinion on cell-type relevance for each software and four diseases. Colours indicate the software used to calculate tissue-type enrichment: Red, *PRSet*; Blue, *MAGMA*; Green, *LDSC*. Dashed line indicates Bonferroni significance threshold of 0.05 for 6 tests (3 methods x 2 models). **E,** Pathway enrichment results for Alzheimer's disease under the top quantile test model. Bar plots show enrichment *P*-values for each cell-type and method. Dashed line corresponds to Bonferroni significance threshold for 22 cell-types (-log10(0.05/22) = 2.64). DOPA, Dopaminergic, Vasc, Vascular; Emb, Embryonic; HP, hypothalamic; Oxt and AVP Exp Neurons, Oxytocin and Vasopressin Expressing Neurons; Nuc, Nucleus; SS, Somatostatin.

reported here should be treated with caution, since they appear highly sensitive to the test model (top quantile / linear) and the number of quantiles used (**Fig 3A and 3D and Fig A-F in S2 Text**). Moreover, there have been several extensions of the *Skene et al* approach, including an extension of MAGMA designed specifically for tissue/cell-type analyses that likely has substantially higher power than the standard MAGMA enrichment tool used here [36]; the basic version of MAGMA as an enrichment tool was used here to enable like-for-like comparisons with PRSet and LDSC regarding power to capture pathway signal.

Our results benchmarking these pathway enrichment tools in multiple settings suggest that PRSet has broadly comparable power to capture genetic signal in pathways as MAGMA and LDSC, with the distinct advantage of providing individual-level estimates of pathway liability, which could be useful in a wide-range of applications. Below, we test the power of pathway PRS for one such application, that of disease stratification.

## Pathway PRSs for disease stratification

While genome-wide PRSs can predict genetic liability to disease because they aggregate individual predictors of disease status, it is unclear if they will be predictive of disease subtypes because they are not optimized to capture disease heterogeneity. In contrast, pathway PRSs may be well suited for disease stratification, since, in theory, the pathway PRS for any pathway that differentiates subtypes can be isolated and exploited for stratification. Given the interest in the potential for PRS to be utilised in stratified medicine [3,8], here we perform a systematic comparison of the predictive power of genome-wide and pathway-based PRSs for subtyping disease.

A common starting point for leveraging PRSs to subtype disease will be one in which: (1) well-powered GWAS data are available only for case-control status, (2) relatively small-sized genotyped samples exist in which subtypes have been identified using e.g. histological, imaging or endoscopic data [37,38], which can be used to train prediction models. These prediction models, ideally based on accessible and cheap information, such as SNP genotypes, can then be used to infer subtypes in large samples without subtype information. Therefore, here we assess the performance of genome-wide and pathway PRSs for disease stratification using a supervised approach that we devised for the purpose, in which polygenic scores are calculated using case/control GWAS effect sizes, and known subtype information is used to optimize the PRS calculation parameters and to train the classification models (**Fig 4A**).

Here we assess the performance of four PRS methods in conducting supervised disease subtyping: (1) PRSet, (2) "PRSet-shift", where gene annotations are shifted by 5Mb to remove their biological meaning (**S3 Text**), acting as a negative matched control to PRSet results, and the genome-wide PRS methods (3) lassosum [27], which is a top-performing PRS method [39] and (4) PRSice [26], which implements the standard C+T PRS method [1] (**Methods**). For (1) and (2), the same 4,079 pathways from existing canonical databases that were used in the previous pathway enrichment section were used to calculate the pathway PRSs. PRSet offers substantially greater modelling flexibility than the two genome-wide PRS methods because it

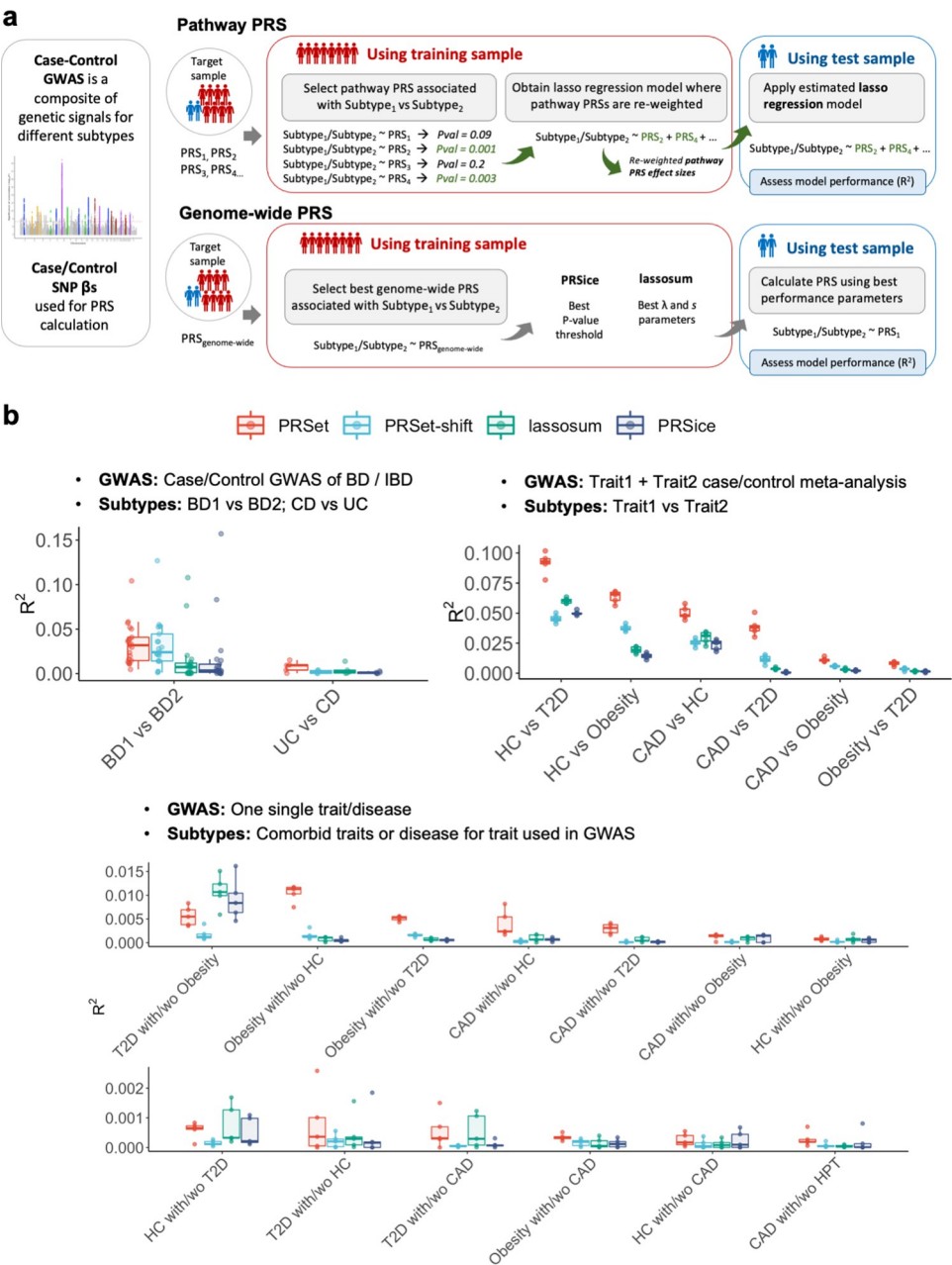

**Fig 4. Stratification of disease subtypes using PRS-based methods using a supervised classification approach. A,** Schematic overview of the pathway PRS and genome-wide PRS approaches for subtype classification. **B, Upper left panel:** Disease stratification of bipolar disorder (BD) and inflammatory bowel disease (IBD) and its subtypes; Bipolar disorder I (BD1), Bipolar disorder II (BD2), Crohn's disease (CD) and ulcerative colitis (UC). **Upper right panel,** Disease stratification of pseudo subtypes of paired major diseases. Subtypes are defined as one major disease vs another. **Lower panel,** Disease stratification of major diseases comorbid subtypes. Subtypes are defined as cases of a major disease with vs without a risk/factor or comorbid trait. T2D, Type 2 Diabetes; CAD, coronary artery disease; obesity (body mass index > 30); HC, hypercholesterolemia (low-density lipoproteins >4.9 mmol/L). HPT hypertension (systolic blood pressure > 140 mm Hg and diastolic blood pressure > 90 mm Hg). Colours indicate the software used to calculate enrichment: Red, PRSet; Light Blue, PRSet with 5Mbp shift; Green, lassosum; Dark blue, PRSice.

optimizes a coefficient for each pathway PRS, while lassosum optimizes only two parameters, and PRSice only one parameter. PRSet-shift offers the same model flexibility as PRSet but with the biological relevance removed and so provides some guide to the predictive boost provided to PRSet by the increased model flexibility alone (**Fig 4B**). Other flexible methods that fit multiple parameters trained to distinguish subtypes can also be developed, as shown in **S3 Text**. However, we did not include these non-PRS approaches in our primary benchmarking since the focus here is on the capacity for PRS-based methods to perform disease stratification, given the intense interest in PRS for stratified medicine [3,8].

We use a range of disease subtype definitions to benchmark the supervised models. First, we use two diseases with well-established subtypes: inflammatory bowel disease and bipolar disorder. Second, we leverage the large number of individuals in UK Biobank with major diseases: type 2 diabetes (N = 19,668), coronary artery disease (N = 22,388), hypercholesterolemia (N = 26,561), and obesity (N = 92,818), to produce composite phenotypes. We combine these outcomes into pairs to mimic a GWAS of a heterogenous disease with two major subtypes, and define each individual disease as a "pseudo subtype". While these pseudo subtypes are unrealistic, assessing the performance of the PRS methods in this setting provides a guide to their relative performance in stratifying real (well-powered) disease data. In the third approach, we define subtypes of coronary artery disease, hypercholesterolemia, hypertension, type 2 diabetes and obesity as the presence/absence of comorbidity within each pair of these diseases (e.g. subtype 1; cases coronary artery disease *with* hypercholesterolemia, subtype 2; cases of coronary artery disease *without* hypercholesterolemia).

**Disease stratification of inflammatory bowel disease and bipolar disorder subtypes.** For the analysis of inflammatory bowel disease, we use publicly available summary statistics for inflammatory bowel disease [40] to calculate PRSs in a sample of UK Biobank participants diagnosed with Crohn's disease (N = 2,101) or Ulcerative colitis (N = 3,681). The UK Biobank sample was then split in *training* (80%) and *test* (20%) samples to optimize and test the stratification models, respectively.

For the analysis of bipolar disorder, we use individual data from 55 cohorts with bipolar disorder case/control status and its subtypes, obtained through collaboration with the Bipolar Disorder working group of the Psychiatric Genomics Consortium [41]. Bipolar disorder case/control GWAS summary statistics for 34 cohorts were meta-analyzed (22,530 cases and 151,450 controls. Effective sample size: 55,862), and the meta-analysis effect sizes were used to calculate PRS for each individual in the remaining 21 cohorts (N = 14,459 individuals with bipolar disorder, of which 10,955 were diagnosed with bipolar disorder I and 3,504 with bipolar disorder II). We perform a leave-one-cohort-out approach to optimize and test the stratification models, where 20 cohorts were used to optimize the PRS and train the classification model, and the remaining cohort was used to validate the model performance.

While the discriminatory power for classifying subtypes was overall low, PRSet outperformed PRSet-shift and the genome-wide PRS methods. The median $R^2$ estimate using PRSet was $9.27 \times 10^{-3}$ for discriminating Crohn's disease vs Ulcerative colitis, and $R^2 = 0.032$ for discriminating Bipolar disorder I vs Bipolar disorder II. For Bipolar Disorder, PRSet-shift and PRSet had comparable performance and both outperformed PRSice and lassosum (**Fig 4B upper left panel and Table F in S1 Tables**). The observation of similar performance between PRSet and PRSet-shift for bipolar disorder is noteworthy, since for most of the other results (see below) PRSet outperforms PRSet-shift substantially and the bipolar disorder analyses are the only ones performed outside of the UK Biobank. The inclusion of such multi-cohort data sets increases heterogeneity, which may reduce the power of our approach since PRSs typically have lower predictive accuracy between rather than within cohorts, and this reduction in accuracy may be critical at the pathway-level. Alternatively, bipolar disorder might be particularly

influenced by genetic variation in regulatory non-coding regions, and so only including SNPs located in coding regions, as in these analyses, would have a limited improvement in the performance of PRSet relative to PRSet-shift.

**Disease stratification of "pseudo subtypes" of paired major diseases.** In the absence of well-established subtypes for type 2 diabetes, coronary artery disease, hypercholesterolemia, and obesity outcomes, we produce "pseudo subtypes" by combining the 5 outcomes into pairs. We meta-analyse the two GWAS of each pair and used the meta-analysis SNP effect sizes in the PRS calculation. We then apply the supervised classification approach as performed for inflammatory bowel disease and bipolar disorder (see **Methods**). In several scenarios, PRSet showed strikingly higher subtyping power than the other methods, suggesting a distinct advantage of the pathway PRS approach in this setting (**Fig 4B** upper right panel and Table G in **S1** Tables).

**Disease stratification for comorbid subtypes of major diseases.** In this subsection, PRSs were calculated using effect sizes from one disease GWAS. For example, PRSs based on coronary artery disease GWAS were used to discriminate between coronary artery disease patients *with* type 2 diabetes vs coronary artery disease patients *without* type 2 diabetes.

Stratification performance estimates for these analyses were lower than for the "pseudo subtypes", with $R^2$ estimates $< 0.016$ (**Fig 4B**, lower panel, Table H in **S1** Tables). In comparisons with relatively high $R^2$ estimates, PRSet outperformed the other three methods, whereas in comparisons with lower discriminatory power ($R^2 < 0.002$) all methods showed similar performance.

## Pathway PRSs for disease prediction

While we hypothesised that pathway PRSs may be particularly well suited to stratification of disease subtypes (**S3 Text**), hence our focus on disease stratification (above), it is also worth evaluating their performance in the standard application of PRS predicting the trait or disease (i.e. case/control status, not subtypes) corresponding to the outcome of the base GWAS. Therefore, to give an initial indication of performance, we assessed pathway and genome-wide PRSs for prediction of the same four traits/diseases that were used for the stratification analyses: type 2 diabetes, coronary artery disease, obesity (defined as body mass index $> 30$) and low-density lipoproteins (LDL) (see **Methods**).

In this standard PRS phenotype prediction setting, the relative improvement in performance for PRSet vs the genome-wide methods was reduced relative to the stratification analyses, and in the cases of obesity and LDL lassosum outperformed PRSet. For the four traits assessed, the phenotypic variance explained by PRSice (C+T method) was the lowest (**S3 Fig**).

## Discussion

Here we introduced a novel, pathway-based, polygenic risk score approach and software tool, PRSet, for performing pathway PRS analyses. We demonstrated that pathway PRSs can capture genetic signal across pathways with similar power as MAGMA and LDSC, with the distinct advantage of providing individual-level estimates of pathway liability. However, we do not presently recommend PRSet as an enrichment tool over these established methods, given its lower power under simulation in small target sample sizes (**Fig 2B**). Genome-wide PRSs derived from large-scale GWAS of heritable traits are typically well-powered for target sample sizes of ~1000 individuals [1], but substantially larger target samples sizes are required to achieve similar power when only a subset of the genome is used (**Fig 2B**). However, the capacity of PRSet to capture significant enrichment of genetic signal at the pathway-level highlights the promise of pathway PRSs as higher-resolution, more biologically interpretable, alternatives to genome-wide PRSs.

Next, we assessed the performance of pathway PRS in an application for which there is broad and substantial hope placed in polygenic risk scores: disease stratification. We found that PRSet often outperformed the genome-wide PRS methods lassosum [27], shown to be a top-performing PRS method [39], and PRSice [26], which implements the standard C+T PRS approach [1], in supervised disease subtyping. The substantially higher performance of PRSet versus the genome-wide PRS methods in a high fraction of the scenarios is noteworthy, given that even markedly different PRS methods typically have similar predictive power [39,42,43]. In **S3 Text**, we investigate the possible reasons for the strong performance of PRSet. Briefly, PRSet likely outperforms genome-wide PRS methods here due to: (i) the prioritisation of variants in genic regions, which have higher heritability [25], and the selection of biological pathways with enriched GWAS signal, demonstrated by the higher performance of PRSet vs PRSet-shift in all scenarios, (ii) the greater modelling flexibility gained by using a large number of (pathway) PRSs for each individual to optimise the prediction model, also observed when the modelling flexibility of lassosum and PRSice is increased (see **S3 Text**), (iii) we hypothesise that PRSet has an advantage over genome-wide PRS methods for subtyping because SNPs that distinguish subtypes will have comparatively lower influence in genome-wide PRS than those affecting all subtypes, while any pathway that differentiates between subtypes will be highly weighted in a pathway PRS prediction model. Thus, standard genome-wide PRSs may be limited-by-design in their application to disease stratification, since they are dominated by variants that affect multiple disease subtypes and their genome-wide aggregation of effects reduces their specificity.

The use of pathway PRSs has two major limitations: (i) pathways are not well-defined and so are likely a weak proxy of biological function, (ii) it is challenging to determine which variants should be linked to each pathway. However, the rapid advances being made in functional genomics, via the integration of increasingly rich resources of multi-omics data, can help to address both issues. For example, future pathway PRSs could be enhanced so that pathways are also defined according to robust differential gene co-expression or protein-protein interaction networks. Moreover, pathways could be annotated using SNP-to-gene linking strategies [44], incorporating regulatory elements outside gene boundaries that are active in tissue and cell-types relevant to the disease under study. While the reliability of pathway definition will continue to be a limitation of this approach [45], if it is ultimately genes and their combined functions that lead to phenotype from genotype, then we propose that pathway-level modelling of disease risk, albeit imperfect, could be a critical tool in the future for research and personalized medicine.

Despite intense interest in the potential of polygenic risk scores to contribute to stratified medicine, ours is the first study to systematically benchmark PRS-based methods for stratification of disease subtypes, finding greater promise for the use of pathway-based PRSs than genome-wide PRSs for supervised stratification. We believe that pathway-based PRSs may offer greater promise in delivering stratified medicine for complex diseases than genome-wide PRSs, which typically aggregate disparate forms of risk into a single number. However, despite promising early results for pathway PRSs reported here, including for both subtyping (**Fig 4**) and standard disease prediction (**S3 Fig**), they have several limitations that need addressing, some of which rely on field-level advances, before their potential can be fully realised. A better understanding of how genetics leads to biological function, and the role of pivotal genes in signalling and mechanistic cascades, will contribute to more reliable definitions of pathways and will provide more accurate and powerful modelling of how multiple genetic liabilities may underlie complex disease.

Our new method and software tool, PRSet, provides a novel approach to computing and analysing polygenic risk scores, motivated by the functional sub-structure of the genome and

the heterogeneity of disease. In contrast to genome-wide PRSs, pathway-based PRSs provide high-resolution information about an individual's genetic risk profile aligned to biological function, and thus have the potential to offer greater insights into disease and a more direct route to precision medicine.

## Methods

### Ethics statement

The UK Biobank study was conducted with the approval of the North-West Research Ethics Committee (ref 16/NW/0274; 21/NW/0157) and all participants gave written consent. This research was conducted using UK Biobank Resource under application number 18177. Samples from the Sweden-Schizophrenia Population-Based cohort were obtained from the database of Genotypes and Phenotypes (Study Accession: phs000473.v2.p2). Samples for the classification of bipolar disorder subtypes were obtained through a secondary analysis approved collaboration with the Psychiatric Genomics Consortium Bipolar Disorder Working Group.

### Participants

**UK Biobank.** UK Biobank is a prospective multi-ethnic cohort of 502,493 participants, aged 40–69 years, initially recruited across the United Kingdom between 2006 and 2010, with follow up since. UK Biobank genetic data used in this study included 488,377 samples and 805,426 SNPs.

Standard quality controls were performed, removing SNPs with genotype missingness > 0.02, minor allele frequency (MAF) < 0.01 and with Hardy Weinberg Equilibrium (HWE) $P$-value < $1x10^{-8}$. We removed all individuals who had withdrawn consent, who had a high degree of missingness or heterozygosity and who had mismatching genetically inferred and self-reported sex as reported by the UK biobank data processing team. We also removed individuals who were not of European ancestry based on a 4-mean clustering on the first two principal components, and related samples with kinship coefficient > 0.044 using a greedy algorithm, since present PRS methods have been shown to have relatively poor portability between global ancestries. A total of 387,392 individuals and 557,369 SNPs remained after quality control.

**Sweden-Schizophrenia Population-Based cohort.** Samples from the Sweden-Schizophrenia Population-Based cohort are a subset of the samples of the Psychiatric Genomics Consortium Schizophrenia Working Group. Data processing and quality controls performed on these data are described elsewhere [46]. A total of 4,834 individuals diagnosed with schizophrenia and 6,128 controls were included.

**Bipolar disorder cohorts.** Samples for the classification of bipolar disorder subtypes were collected in Europe, North America and Australia, and included a total of 39,712 individuals with a lifetime diagnosis of bipolar disorder and 178,749 controls. We obtained access to summary statistics for individual cohort case/control GWAS for 55 cohorts, and to individual-level data for 43 cohorts. Imputation, cohort harmonization and quality controls are described elsewhere [41]. Processed and harmonized genotype and phenotype data was used in our study.

### Definition of pathways

KEGG [20], BioCarta [29], Pathway Interaction Database (PID) [30] and Reactome [19] canonical pathways were obtained from the Molecular Signatures Database (MsigDB v7.0) [47]. Pathways from the Gene Ontology database (GO, accessed on 2021-03-17) [32,33] and

Mouse Genome Database (MGD, accessed on 2021-03-17) [31] were also included. For MGD pathways, we i) used the human-mouse homolog list provided by MGD to convert the mouse gene names to their human counterpart and ii) restricted our analyses to pathways with ontology level > 4 to avoid inclusion of pathways that are extremely specific. We removed pathways with fewer than 10 genes or more than 2000 genes to exclude over specific or too broad pathways. A total of 4,079 pathways across the six pathway database resources were included in the analyses.

### Estimation of pathway enrichment

**Definition of phenotypes.** In order to optimise statistical power for benchmarking the performance of the methods tested in the study, we selected complex phenotypes with high SNP-heritability estimates, with publicly available summary statistics from large GWASs and that were measured in UK Biobank or the Sweden-Schizophrenia Population-Based cohort (**Table I in S1 Tables**). As such, we extracted data from UK Biobank on the following phenotypes: body mass index, low-density lipoproteins, coronary artery disease, alcohol consumption, type 2 diabetes, and a proxy of Alzheimer's disease based on parental history of the disease (**S1 Methods**). Schizophrenia cases and controls were extracted from the Sweden-Schizophrenia Population-Based cohort.

**GWAS data sets.** GWAS data sets for body mass index [48], low-density lipoproteins [49], Alzheimer's disease [50], coronary artery disease [51], type 2 diabetes [52] and alcohol consumption [53] were downloaded from public online databases and used without modification. Since the Sweden-Schizophrenia Population-Based cohort was included in the PGC schizophrenia GWAS, we used a version of the GWAS with the Sweden-Schizophrenia cohort excluded [46] to avoid sample overlap and prevent inflation of results.

**Pathway enrichment analyses.** *PRSet*. Pathway specific PRS analyses were performed using PRSice-2 (v2.3.5) on genotype data. The Major histocompatibility complex region (MHC, chr6:25Mb-34Mb) was removed for all the diseases assessed and the APOE region (chr19:44Mb-46Mb) was also removed for Alzheimer's disease. SNPs were annotated to genes and pathways based on GTF files obtained from ENSEMBL (GRCh37.75). We extended the gene coordinates 35 kilobases (kb) upstream and 10 kb downstream of each gene to include potential regulatory elements, but SNPs outside those gene window-boundaries were not included in the PRS. Ambiguous SNPs (A/T and G/C) and SNPs not present in both GWAS summary statistics and genotype data were excluded. 10,000 permutations were performed to obtain empirical "competitive" *P*-values, which account for the number of SNPs included in a given pathway.

PRSet calculates the competitive *P*-values as follows; first, a "background" pathway containing all genic SNPs is constructed, and clumping is performed within this pathway. For pathways with *m* SNPs, *N* null pathways are generated by randomly selecting *m* "independent" SNPs from the "background" pathway. The competitive *P*-value can then be calculated as

$$competitive \ P - value = \frac{\sum_{n=1}^{N} I(P_n < P_o) + 1}{N + 1}$$

where $I(.)$ is an indicator function, taking a value of 1 if the association *P*-value of the observed gene set ($P_0$) is larger than the one obtained from the $n^{th}$ null set ($P_n$), and 0 otherwise. A pseudo-count of 1 is added to the numerator and denominator to avoid competitive *P*-values of 0 and conservatively counting the observed gene set as 1 potential null set [54]. One consideration of this permutation procedure is that the smallest achievable competitive *P*-value is 1/($N$+1), which can lead to difficulties in ranking highly significant gene sets.

*MAGMA*. MAGMA is a software for pathway enrichment analysis using GWAS data. The implementation of MAGMA can be divided in two parts: a gene level analysis and a pathway level analysis. First, the gene level analysis is performed by combining the GWAS *P*-values of SNPs around a gene (for GWAS data) or genotype data (when this is available at the individual level|) to compute a gene test statistic. This gene level analysis takes into account LD structure by using a reference data set.

For the pathway analysis, the gene level association statistics are transformed to Z-scores. These Z-scores reflect how strongly each gene is associated with the phenotype, with higher values corresponding to stronger associations. MAGMA has a competitive pathway analysis test that is calculated as:

$$Z = \beta_0 + I\beta_p + C\beta_c + \varepsilon$$

where I is an indicator variable that takes the value of 1 if a gene is included in pathway *p*, or the value of 0 if gene g is not in pathway *p*, and C is a matrix of covariates. The *P*-value results from a test on the coefficient $\beta_p$, which assesses whether the phenotype is more strongly associated with genes included in a pathway than with genes not included in the pathway.

To directly compare the performance of PRSet vs MAGMA (v1.07b) given identical input data, we removed all ambiguous SNPs and non-overlapping SNPs prior to MAGMA analyses. It is important to note that this step is unnecessary for MAGMA and might negatively impact its performance. After filtering, gene-based analyses were performed on GWAS summary statistics using the '—*pval*'function, and genotype data for the target samples independently. As in PRSet analyses, a 35kb window upstream and a 10kb window downstream were added to gene coordinates, the MHC region was excluded for all traits, and the APOE region was excluded for Alzheimer's disease. Gene-based results were then meta-analysed using the inbuilt '—*meta*'function and were subsequently used as input to the pathway analysis.

*LDSC*. The LDSC method relies on the fact that in GWAS the $\chi^2$ association of $SNP_i$ with a phenotype includes the effects of all the SNPs tagged by $SNP_i$. This means that for polygenic traits (where small genetic effects are spread across the genome) the strength of the relationship between each SNP $\chi^2$ and the trait should be proportional to the heritability the SNP tags [55]. LDSC requires only GWAS summary statistics and LD information from an external reference panel that matches the population studied in the GWAS.

Stratified LDSC is an extension of the original LDSC method that partitions heritability from GWAS summary statistics into functional categories (e.g. pathways) [25]. The resulting partitions, called partitioned LD scores, are then used to estimate the enrichment in heritability for each category. Heritability enrichment is defined as the proportion of SNP-heritability captured in a functional category divided by the proportion of SNPs in that category. To estimate the SNP-heritability, heritability for each SNP ($\tau_c$) is estimated via multiple regression while accounting for LD, sample size and other confounding biases. It assumes that under a polygenic model the expected $\chi^2$ of $SNP_i$ is

$$E[\chi_i^2] = N\sum_C \tau_c \ell(j, C) + Na + 1$$

where N is sample size, C indexes categories, $\ell(j, C)$ is the LD score of $SNP_i$ with respect to category C, and a is a term that measures the contribution of confounding biases. If the functional categories are disjoint, $\tau_c$ is the per-SNP heritability in category C. If categories overlap, the per-SNP heritability is the sum of the SNP-heritability across categories ($\sum_{C:i \, \epsilon \, C}\tau_c$).

Partitioned LD scores were calculated using the 1000 Genomes European genotype data as reference panel [56]. Similar to PRSet and MAGMA, SNPs were annotated to genes and pathways with 35kb upstream and 10kb downstream extension prior to calculation of LD scores.

Ambiguous SNPs and non-overlapping SNPs were removed prior to LDSC analyses to allow for direct comparison between PRSet and LDSC. GWAS were performed on the target genotype data using PLINK v1.90b6.7 [57], and were meta-analysed with the external GWAS summary statistics using METAL (2011-03-25) [58]. Partitioned LD score regression was then performed using LDSC v1.01 [25,55], with the MHC (all traits) and APOE (Alzheimer's disease only) regions excluded.

## Evaluation of pathway enrichment using canonical pathway definitions

**Assessment of pathway enrichment by simulation.** *Generation of causal pathways.* Out of 4,079 empirical pathways extracted from six publicly available collections (see "definition of pathways" section), we randomly selected 50 or 4,050 pathways and defined them as 'causal'. Each of the 'causal' pathways was randomly assigned with a certain level of enrichment, ranging from 1 to 30%, with step size of 1%. This means that for each pathway, we selected between 1 and 30% of the SNPs included in the pathway and added them to a list of 'causal SNPs'. This list of SNPs was then used to assess pathway enrichment for each of the 4,079 empirical pathways and rank them based on their enrichment (**S4 Fig**). The simulation process was repeated 20 times.

*Phenotype simulation and sample selection.* Simulation was performed using UKB genotype data. Quantitative traits ($Y$) with SNP-based heritability ($h^2$) of 0.1 or 0.5 were simulated as $Y = X\beta + \varepsilon$, where $X$ is the standardized genotype matrix, $\varepsilon$ is the random error defined as $\varepsilon \sim N(mean = 0, sd = \sqrt{var(X\beta)(1 - h^2)})$, and $\beta$ is a vector of SNP effect sizes which follows a point-normal distribution $\beta \sim N(mean = 0, sd = \sqrt{h^2})$, with non-causal SNPs assigned with $\beta = 0$.

For each trait, 50$k$, 125$k$ or 250$k$ individuals from European ancestry were randomly selected to generate the GWAS summary statistics using PLINK v1.90.b6.7. An independent set of either 1$k$, 10$k$ or 100$k$ individuals were then randomly selected as the target samples. Pathway analyses were performed as described in the previous sections.

Agreement between pathway enrichment results for PRSet, MAGMA and LDSC and the rank of empirical pathways was assessed by calculating the Kendall correlations between the -log10 competitive $P$-value generated by each pathway enrichment tool, and the ranks of pathways based on enrichment of simulated causal variants.

**Assessment of pathway enrichment using *MalaCards* relevance scores.** To assess whether pathway enrichment results were in line with previous biological knowledge on the phenotypes of interest, disease-associated relevance scores for each pathway were constructed using information from the *MalaCards* database [59]. The *MalaCards* database provides a disease relevance score *for each gene* based on experimental evidence and co-citation in the literature. For the six diseases included in this analysis (schizophrenia, Alzheimer's disease, alcohol consumption, low-density lipoproteins, coronary artery disease and body mass index), we downloaded the *MalaCards* disease-associated relevance scores (Accessed on 2020-11-27, see **Table J in S1 Tables** for disease terms used and number of genes). Next, we performed a rank normalization of the scores where, assuming that a disease has $n$ genes with *MalaCards* scores, a score of $(r+1)/(n+1)$ were assigned to each gene, with r being the inverse ranking of the gene with *MalaCards* score. Genes without a *MalaCards* score are assigned a score of 0. *MalaCards* provide gene information as gene symbols, which were transformed to ENSEMBL gene names.

Since MalaCards scores only relate to genes, we computed disease-associated relevance scores *for each pathway*. We calculated the sum of the rank transformed *MalaCards* scores for the genes included in a pathway and divided by the number of genes in the pathway to account for pathway size (**S5 Fig**).

Agreement between pathway enrichment results for PRSet, MAGMA and LDSC and the *MalaCards disease relevance scores* was assessed by calculating the Kendall correlations between the -log10 competitive *P*-value generated by each pathway enrichment tool, and the *MalaCards* relevance score for each pathway.

### Evaluation of pathway enrichment using tissue/cell-type defined pathways

**Defining tissue specificity sets from bulk-tissue RNA-sequencing data.** To calculate tissue specificity across pathways, we obtained bulk-tissue RNA-sequencing gene expression data from 55 tissues from the GTEx consortium [60] (v8, median across samples). Tissues with less than 100 individuals, cancer related tissue types (e.g. EBV-transformed lymphocytes and Leukemia cell line), and testis (which were considered as an outlier [61]) were removed, retaining a total of 47 tissues. We filtered out all non-protein-coding genes and genes not expressed in any tissue.

Gene expression specificity was calculated by dividing the expression of each gene by its total expression across tissues [61]. The resulting gene expression specificity ranged from 0 (gene is not expressed) to 1 (gene is exclusively expressed in this tissue). Next, expression specificity of each tissue was divided into 11 quantiles following the approach introduced in *Skene et al 2018* [34], where the first quantile contained all non-expressed genes in a given tissue, and the 11[th] quantile contained the most specifically expressed genes. Genes within each quantile were grouped into a single pathway.

**Defining the cell-type specificity sets.** Cell-type specificity data were obtained from supplementary materials of Skene et al (2018) [34] which includes gene expression specificity information for 24 brain cell-types obtained from single cell RNA-sequencing data. Again, expression specificity of each brain cell-type was divided into 11 quantiles with the first quantile containing all non-expressed genes in a given cell-type. Genes within each quantile were grouped into a single pathway.

**Ranking the importance of cell-type / tissue.** To provide an objective estimate of tissue / cell-type importance for each phenotype, we invited two experts (per phenotype) who were blind to our experiment and algorithm design to provide their opinion on what cell-type(s) and tissue(s) are expected to be implicated for each disease context (**Table C in S1 Tables**). The expert response was coded as "none" (both experts think tissue/cell-type is not implicated), "single" (only one expert thinks a tissue/cell-type is important) and "both" (both experts agree about the importance of a tissue/cell-type).

**Cell-type and tissue specificity analyses.** We used two testing strategies to assess the relationship between disease GWAS signals and tissue(s)/cell-type(s) specificity (**S6 Fig**). For the *Top quantile enrichment strategy* GWAS signals are enriched in the most specifically expressed genes [34]; whereas for the *Linear enrichment strategy* GWAS signals increase linearly with expression specificity [24,36]. The top quantile strategy reports the competitive *P*-value of the pathway defined by those genes in the top expression specificity quantile for each software and tissue/cell-type. The linear enrichment strategy fits a linear regression with the -log10 competitive *P*-value for each of the pathways defined by the expression specificity quantiles as dependent variable, and the quantile ranks as the predictor variable, and reports the one-sided *P*-value for a positive association.

The concurrence of the methods' ranking of the tissues / cell-types with that of the experts within each disease for both the top quantile and linear enrichment strategies was measured by regressing the inverse normalized -log10 *P*-value for the top quantile / linear enrichment strategies for each cell-type / tissue against the expert opinion, coded as factor.

MAGMA has a specific model which accounts for expression specificity ('—*gene-covar*'). However, in favour of a more consistent analysis between the three software methods, this

model was not used. It is thus possible that MAGMA can provide more powerful results using the dedicated model.

Results from the regressions against the expert confidence score assessed the association of the gene expression specificity and GWAS signal with the expert opinion for each pathway enrichment software under each of the two hypotheses.

### Disease stratification

**Description of GWAS and target datasets.**  *Inflammatory bowel disease subtypes*. As base sample, we used publicly available summary statistics from a case/control inflammatory bowel disease GWAS [40]. The SNP effect sizes of this GWAS were used to calculate pathway and genome-wide PRS for each individual in the target sample, composed by UK Biobank participants diagnosed with Crohn's disease and with ulcerative colitis. The target sample phenotype was encoded as individuals with Crohn's disease *vs* individuals with ulcerative colitis.

*Bipolar disorder subtypes*. We obtained access to individual genotype data from 55 bipolar disorder cohorts collected by the PGC Bipolar Disorder Working group (**Table K in S1 Tables**). Quality control, imputation and harmonisation was performed on this data as previously described [41]. Out of the 55 cohorts, we selected 34 as base sample and meta-analysed each cohort case/control GWAS results using the software METAL (2011-03-25) [58] with the sample-size weighted fixed-effects algorithm. We used the remaining 21 cohorts as target sample and calculated for each individual with bipolar disorder pathway and genome-wide PRS. The target sample phenotype was encoded as individuals with bipolar disorder I *vs* bipolar disorder II.

*Pseudo subtypes of paired major diseases*. We obtained previously published GWAS summary statistics for four major diseases: type 2 diabetes, coronary artery disease, obesity (defined as body mass index > 30) and hypercholesterolemia (defined as low-density lipoproteins > 4.9 mmol/L) and performed a meta-analysis for each pair of traits. Meta-analyses were performed using METAL [58] with the sample-size weighted fixed-effects algorithm. To truly mimic a composite phenotype GWAS, only variants included in both GWAS summary statistics were retained. The resulting meta-analysis summary statistics were used as base sample. As target sample, we generated composite phenotypes by combining cases of the two paired phenotypes using UK Biobank. To calculate the PRS, target sample phenotypes were encoded mimicking sub-phenotypes of a given disease, for example, for the phenotype coronary artery disease-obesity, samples with coronary artery disease (and not obesity) were coded as 0 and those with obesity (and not coronary artery disease) were coded as 1 (**Tables L-N in S1 Tables**).

*Comorbid subtypes of major diseases*. For the analysis of subtypes with presence/absence of comorbid diseases, we used type 2 diabetes, coronary artery disease, obesity, hypertension and hypercholesterolemia, as these diseases present high comorbidity between them (**Tables L-N in S1 Tables**). As base sample, we used publicly available GWAS summary statistics for one of the diseases (e.g. type 2 diabetes). As target sample phenotypes, we defined subtypes of a disease as the presence/absence of the other disorders (e.g. type 2 diabetes with obesity vs type 2 diabetes without obesity).

**Target sample split for cross validation and leave one cohort out analyses.**  For the optimization of PRS and stratification steps using UK Biobank data, we performed a 5-fold cross validation approach. For each fold, the target sample was randomly split into a training (80% of target) to optimize the PRS and lasso regression parameters, and a test sample (20% of target) to assess out-of-sample method performance.

For the analysis of Bipolar Disorder, we performed a leave-one cohort out approach to maximize the sample size used for optimizing PRS and lasso regression parameters. Out of the 21

cohorts selected as target sample, we used 20 cohorts to optimize the stratification (training cohorts), and the remaining cohort was used to test the method performance.

**Calculation and optimization of PRSs using the training sample.** For the phenotypes ascertained using UK Biobank, sex, age, age of diagnosis (for coronary artery disease and type 2 diabetes), genotyping batch, recruitment centre and first 15 principal components were adjusted using logistic regression analyses. For bipolar disorder, the first five principal components and any others required for each cohort were adjusted for using logistic regression. For all phenotypes pseudo residuals obtained from the logistic regressions were used as the outcome variable in PRS analyses.

Pathway-specific PRSs for 4,079 pathways were calculated using PRSet. Competitive $P$-values were calculated using 10,000 permutations and pathways with competitive $P$-value $< 0.05$ were defined as enriched (see definition of pathways and pathway enrichment sections). PRSs for the enriched pathways were recalculated using $P$-value thresholding, such that the predictive power of each PRS was maximized. We also performed genome-wide PRS analyses using lassosum and PRSice-2. Optimal parameters for the training sample phenotype prediction ($P$-value thresholds for PRSice; penalty factor λ and soft-thresholding parameter $s$ for lassosum) were extracted. All PRSs were standardised to have mean 0 and standard deviation of 1.

**Supervised analyses for classification of disease subtypes.** *Supervised classification using pathway PRSs*. Enriched pathway PRSs (with competitive $P$-value $< 0.05$, obtained after running PRSet with $P$-value threshold of 1) at their "best" predictive $P$-value threshold were included in a generalized linear model with lasso regularization using the '*cv.glmnet*'function from the glmnet package (v4.0–2) in R. '*cv.glmnet*' takes as input (1) a matrix with PRSs for each individual and each pathway, where rows correspond to individuals in training sample size and columns correspond to the number of enriched pathway PRSs, and (2) the subtype information for each individual. We performed a 5-fold cross-validation to select the lasso lambda parameter that generates the smallest out-of-sample mean squared error (MSE). By using a lasso regularization approach, we remove redundant signal between enriched pathways and re-adjust the effect size of the PRSs to optimize subtype classification (Note that all PRSs were calculated using case-control GWAS effect sizes). The resultant best fitting glmnet model was then applied to the test sample using the '*predict*' function also included in the glmnet package. The predicted values were compared with the known subtype information in the test sample to calculate the model $R^2$.

*Supervised classification using genome-wide PRSs*. Genome-wide PRS with the best $P$-value threshold (for PRSice) and best λ and $s$ parameters (for lassosum) obtained using the training sample were applied to calculate PRS for the test sample and to calculate the model $R^2$.

## Single trait prediction

Genome-wide and pathway specific PRS were calculated for the same four phenotypes that were used for the classification of subtypes: type 2 diabetes, coronary artery disease, obesity (defined as body mass index $> 30$) and low density lipoproteins. We calculated PRS for these traits using publicly available GWAS data for individuals from UK Biobank cohort as described for classification of disease subtypes.

We then performed a supervised classification using pathway PRS, where we selected enriched pathway PRS (competitive $P$-value $< 0.05$) at their best predictive $P$-value threshold, and included them in a generalized linear model with lasso regularization using the '*cv. glmnet*'function. In this case, the '*cv.glmnet' function* takes as input (1) a matrix with PRS for each individual and each pathway and (2) the case/control information for each individual (Instead of the subtype information for each individual used in the classification of subtypes section). The resultant best fitting glmnet model was applied to the test sample.

We applied the standard procedure or the prediction of single traits using genome-wide PRS. The PRS with the best *P*-value threshold (for PRSice) and best λ and *s* parameters (for lassosum) were obtained using the training sample and applied on the test sample to calculate the model $R^2$.

## Supporting information

**S1 Acknowledgements. Bipolar Disorder Working group of the Psychiatric Genomics Consortium list of collaborators.**
(DOCX)

**S1 Methods. Supplementary methods.**
(DOCX)

**S1 Tables. Supplementary tables.** Table A. Kendall correlation coefficients (τ) between pathway ranks based on competitive P-values of enrichment computed by each software and the empirical pathway ranks based on the true (simulated) effects across the pathways. Table B. Kendall correlation coefficients (τ) between pathway ranks based on competitive P-values of enrichment computed by each software and pathway ranks based on MalaCards disease relevance scores. Table C. Expert opinion on tissue and cell type relevance for each disease. Table D. Pathway enrichment results. Table E. Association between pathway enrichment P-value for each software and six diseases and expert opinion of tissue and cell type relevance. Table F. Stratification of inflammatory bowel disease and Bipolar Disorder subtypes. Table G. Stratification of "pseudo subtypes" of paired major diseases. Table H. Stratification of comorbid subtypes. Table I. Cohorts used as base and target samples in analyses evaluating pathway enrichment (Post genetic QC). Table J. Phenotypes used in pathway enrichment analyses and correlation with Malacards relevance scores. Table K. Bipolar Disorder cohorts used for classification of bipolar disorder subtypes. Table L. GWAS summary statistic used in the meta-analysis for sub-phenotype classification analyses. Table M. UK Biobank samples used in analyses using composite diseases / traits. Table N. Coding correspond to statin in UK Biobank medication records (Field ID 20003). Table O. S1 Table References.
(XLSX)

**S1 Text. Sensitivity analysis excluding genes in *MalaCards* database.**
(DOCX)

**S2 Text. Pathway enrichment results for Pathways defined using tissue/cell-type specificity.**
(DOCX)

**S3 Text. Evaluating and discussing the mechanisms underlying PRSet performance for the classification of disease subtypes.**
(DOCX)

**S1 Fig. Illustration of bit operation that helps to optimize PRSet clumping.** The index SNP will "remove" gene set memberships from the clumped SNPs if and only if they fall within the same gene set. Clumped SNP without any gene set membership will be removed at the end of clumping. Here, clumped SNP 2 will be removed.
(TIF)

**S2 Fig. Additional results for evaluating the power of PRSet using a pathway enrichment approach. a)** Simulation analyses– 4050 pathways. Performance was defined as the Kendall correlation between the competitive *P-value* for each software and the empirical pathway

ranking. Boxplots illustrate the values of Kendall rank correlation coefficients ($\tau$) for PRSet, MAGMA and LDSC for each combination of heritability ($h^2$ = 0.1, 0.5) base sample size used in GWAS n = (50K, 125K, 250K), and target sample size n = (1K, 10K, 100K). **b)** Kendall correlation coefficients ($\tau$) between pathway enrichment analyses and *MalaCards* relevance scores. Bar plots illustrate joint results of the six databases used to define pathways. *empirical *P-value* < 0.05.
(TIF)

**S3 Fig. Performance of PRSet vs genome-wide PRS methods for prediction of single traits.** CAD, coronary artery disease; HC, hypercholesterolemia; T2D, type 2 diabetes disease.
(TIF)

**S4 Fig. Flowchart depicting the generation of 50 simulated causal pathways and pathway ranking.** The same approach was used for the simulation of 4,050 causal pathways.
(TIF)

**S5 Fig. Flowchart depicting generation of pathway based *MalaCards* scores.**
(TIF)

**S6 Fig. Illustration of the test models used to assess cell type and tissue specificity.** Left panel: illustrates the "top quantile" test model, which assumes that GWAS signal enrichment is concentrated in the most specifically expressed genes. Right panel: illustrates the "linear" test model, which assumes that enrichment of GWAS signal increases linearly with expression specificity.
(TIF)

## Acknowledgments

We thank the participants in UK Biobank and the scientists involved in the construction of this resource. We thank Dr Kristen Brennand, Dr Jason Kovacic, Professor Alison Goate, Professor Ruth Loos, Dr Edoardo Marcora, Dr Alexander Charney, Dr Manav Kapoor and Dr Jacqueline Meyers for providing their expert knowledge for each specific disease. We thank Dr Conrad Iyegbe, Laura Sloofman, Collin Spencer, Dr Zhe Wang and Dr Jiayi Xu for useful discussions and feedback. Fig 1 was partially created using the resource BioRender.com.

## Author Contributions

**Conceptualization:** Shing Wan Choi, Judit García-González, Paul F. O'Reilly.

**Data curation:** Shing Wan Choi, Judit García-González, Hei Man Wu, Jessica Johnson.

**Formal analysis:** Shing Wan Choi, Judit García-González, Yunfeng Ruan, Hei Man Wu.

**Funding acquisition:** Hei Man Wu, Paul F. O'Reilly.

**Investigation:** Shing Wan Choi, Judit García-González, Yunfeng Ruan.

**Methodology:** Shing Wan Choi, Judit García-González, Yunfeng Ruan, Hei Man Wu, Clive J. Hoggart, Paul F. O'Reilly.

**Project administration:** Paul F. O'Reilly.

**Software:** Shing Wan Choi, Judit García-González, Yunfeng Ruan.

**Supervision:** Clive J. Hoggart, Paul F. O'Reilly.

**Validation:** Shing Wan Choi, Judit García-González, Christian Porras.

**Visualization:** Shing Wan Choi, Judit García-González.

**Writing – original draft:** Shing Wan Choi, Judit García-González, Hei Man Wu, Paul F. O'Reilly.

**Writing – review & editing:** Shing Wan Choi, Judit García-González, Yunfeng Ruan, Hei Man Wu, Christian Porras, Jessica Johnson, Clive J. Hoggart, Paul F. O'Reilly.

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
