## [Decision Letter · Decision Letter 0]

14 Jun 2022

Dear Dr García-González,

Thank you very much for submitting your Methods entitled 'PRSet: a tool for pathway-based polygenic risk score analyses' to PLOS Genetics.

The manuscript was fully evaluated at the editorial level and by independent peer reviewers. The reviewers appreciated the attention to an important problem, but raised some substantial concerns about the current manuscript. Based on the reviews, we will not be able to accept this version of the manuscript, but we would be willing to review a much-revised version. We cannot, of course, promise publication at that time.

If you decide to revise the manuscript for further consideration at PLOS Genetics, please aim to resubmit within the next 60 days, unless it will take extra time to address the concerns of the reviewers, in which case we would appreciate an expected resubmission date by email to plosgenetics@plos.org.

[LINK]

We are sorry that we cannot be more positive about your manuscript at this stage. Please do not hesitate to contact us if you have any concerns or questions.

Yours sincerely,

Heather J Cordell

Associate Editor

PLOS Genetics

Xiaofeng Zhu

Section Editor: Methods

PLOS Genetics

Reviewer's Responses to Questions

**Comments to the Authors:**

Reviewer #1: This manuscript presents a novel method that allows splitting of polygenic risk scores (PRS) into subsets according to externally-defined functional pathways. These pathway-specific PRS can then be used to investigate pathway enrichment, distinguish subtypes of disease or be applied to the prediction of single traits.

Overall the investigations of performance of the various methods do not seem sufficiently wide-ranging and are often unrealistic, particularly for the application of PRS to disease stratification. the methods section is unclear in places.

I've presented my comments in the order that the relevant sections occur in the manuscript. This has the result that some comments on similar aspects of the methodology appear early on (for the main manuscript 'Results' section) and some later (for the 'Methods' section).

p.1 "We find that pathway PRSs have similar power for evaluating pathway enrichment of GWAS signal as leading methods MAGMA and LD score regression". I'm not convinced your simulations support this. PRSet is the least powered for a target dataset of 1000, less well-powered than MAGMA for a dataset of 10,000 and only marginally better than MAGMA for a sample size of 100,000.

p.1 "Using UK Biobank data, we show that pathway PRSs can outperform genome-wide PRSs for trait prediction and stratification of diseases into subtypes". while it "can" this is a bit vague. Other methods "can" outperform pathway PRS.

p.5 It seems odd to start with pathway enrichment, given that the principal reason for developing PRS is risk prediction/discrimination.

p.6 "GWAS were then performed on 250k individuals and their simulated traits". Why only one size of training data but multiple sizes of testing data? For a simulation investigating a method of this sort I would expect a broader investigation of scenarios.

p.6 Given that you only ran 20 simulations are differences in the Median Kendall values reliable? Why so few simulations? How different are your results if you choose a different 50 pathways?

p.6 The simulations (here and in later sections) assume that the pathways are predicted without error. What if a proportion of them are incorrect (as is likely)? How will this affect the results? And which methods are most robust to this? Similarly what happens if SNPs lie in multiple pathways - it's unclear how this is handled. What about SNPs that are highly significant but aren't predicted to lie in any pathway? This is of relevance here and in other sections but is only mentioned in passing.

p.10 "Pathway PRSs for disease stratification" - why are there no simulations for this? This could be done to compare two theoretical traits with different degrees of genetic correlation and heritability. Without this it's hard to ascertain when one method might be expected to outperform another especially since the 'real' data is mostly unrealistic for this scenario (see below).

p.13 The 'pseudo-subtypes' seem artificial and unrealistically different. You say these are "mimicking a GWAS on a heterogenous disease with major subtypes", but traits like 'extreme height' and 'type II diabetes' will have far less genetic correlation than real disease subtypes, so this seems an unfair test of performance. I would have liked to have seen more diseases/subtypes for which PRS might be expected to be useful in terms of having similar symptoms. Why not different autoimmune disorders, types of cardiovascular disease, closely-related cancers, psychiatric disorders etc. where there are known genetic differences already but genetic correlations are high? Even if numbers are small in UK Biobank summary stats should be available from consortia and UK Biobank could be used as a testing set.

p.16 "Pathway PRSs for single trait prediction" - this is a surprisingly brief section (just 13 lines), particular given the potential importance of including pathway modelling in PRS. For example why no inclusion of MegaPRS (https://www.nature.com/articles/s41467-021-24485-y) here or elsewhere given its claims to substantially outperform all existing PRS methods?

p.20 The Sweden-Schizophrenia Population-Based cohort is mentioned but only the QC for UK Biobank is described. Moreover overlap with UK Biobank samples is described as an issue. So was the Swedish dataset used as a training dataset and UK Biobank as a testing set? This needs to be explained both in terms of the analyses conducted and QC undertaken.

p.25 You don't say in the Methods that you only include SNPs that are in genes in the pathway being considered. Presumably this is the case but it should be stated. Do the pathway-specific PRS only contain SNPs in or near genes? What happens to the other SNPs that would otherwise reach the p-value threshold for inclusion? Is the pathway-specific PRS biased towards bigger genes with more SNPs while the 'background' pathway (used to generate the null) lacks that bias (given that in your simulations SNPs are randomly assigned)? Have you checked the p-value under the null - none of your simulations for using PRS to distinguish traits look at p-values so it's hard to know whether there is bias?

p.25 Why does the heritability model not depend on the number of SNPs? how is the number of SNPs in the model determined?

p.31/32 Each pathway-specific PRS is optimised in the training set and then the lambda value to weight each of these PRS seems to be optimised in the same dataset using Lasso. So how much does this differ from using a Lasso model to create a PRS in the first place? Are pathways narrowed down to those of relevance to the disease or those that include the most significant SNPs? If not (given that you mention 4,079 pathways) don't you end up with a lot of irrelevant pathways and so redundant pathway-specific PRS which leads to an unnecessary multiple testing burden? I'm unconvinced by the potential gain here having a two-step process in the same training set. In the 'supervised' analysis are all steps in PRS creation aimed at distinguishing the subtypes or do some use the overall case-control definition?

p.32/33 "SNP-Stratifier method for classification" - you say you use "post-clumped" SNPs and then re-estimate effect sizes/weights using a Lasso method for case-case status. So it sounds like one SNP per region is selected based on the most significant from the case-control analysis and then the effect re-estimated based on the case-case status. But the most significant SNP for a case-case comparison may not be the same as the most significant for the case-control analysis. Can you clarify? And, if I understand correctly, would you not be better not doing the clumping (to thin SNPs) but just using Lasso regression to pick the best SNPs for the case-case comparison?

Minor:

There are a lot of places where there are grammatical errors, typos or the English is just unclear:

p.6 "being best-performing method for 100k target data" should be "being the best-performing method for the 100k target data"

p.22 "It does this combining the GWAS P-values of SNPs" should be "It does this by combining the GWAS P-values of SNPs"

p.23 I don't understadn this sentence: "βp is the difference of association of genes in the pathway with phenotype and the association of genes outside the pathway with the phenotype"

p.23 Similarly "The competitive tests the null hypothesis"

p.23 "Same as for PRSet analyses" should read soemthing like "As in PRSet analyses"

The authors repeatedly refer to "the UK Biobank" but I think it should be just "UK Biobank"

p.32 "We used PRSice `--print-snp` command" should be "We used the PRSice `--print-snp` command"

Reviewer #2: Choi, O'Reilly and colleagues present a novel approach that leverages the PRS toolkit to deliver pathway based analysis. The aim is laudable, and it is very clear that being able to de-convolute a genome-wide PRS into biologically interpretable components is potentially extremely valuable. I can see plenty of applications, especially for targeted interventions to understand the pathways most at risk in given individuals. However, while I am excited about the aim, I really struggle to understand the technicalities of the paper.

A key issue for me is the concept of test set. As far as I understand methods like MAGMA and LDSC, these only take as input the GWAS data (but perhaps I am misunderstood?). I cannot see how the test set would impact the performance of these methodologies, that should really be driven only the power of the underlying GWAS study. I see that a test set is useful for PRSSet, because of the way the PRS must be deployed in a dataset with individual level data. But with that in mind, Figure 2A confuses me quite a bit, given my understanding of MAGMA and LDSC. Perhaps this is consistent with the performances that do not vary with the size of the test set. But that tells me that this evaluation process is a bit odd.

I also have some (less serious) difficulties with the simulation study for pathway enrichment. I understand that variants were selected as causal within each pathway, but are the authors really assuming that 5-50% of SNPs in a pathway are causal? That does seem very large. The evaluation process also seems quite complex: (i) generate a P-value for each pathway, (ii) compare that P-value to the null by generating P-values for random pathways of the same size, resulting in a competitive P-value for that pathway (iii) compare the competitive P-value significance ranks across pathways between simulation and truth to generate a correlation score. Did I get this right? If so, some visual to guide the reader in that process would really be helpful as it took me multiple reads and I am still unsure.

The section on MalaCards relevance scores was also hard to follow. The process to go from disease/gene specific scores to pathway based rankings does seem quite arbitrary. I do not have a particular issue with the process, but I would like to understand how "canonical" that process is. And also see some visual to support the reader.

Following a similar theme, the disease stratification work (supervised or unsupervised) is hard for me to parse. I am not sure how the optimisation is performed using the test set. The methods section refers to "linear regression models", but it probably should be explained in greater details. The expectation that an unsupervised strategy may be sufficient to separate cases of Crohn and UC seems quite unrealistic given how similar these diseases are genetically, hence I would simply remove that unsupervised section that adds little to the paper.

A suggestion on disease stratification analysis, perhaps off topic but of interest to me: I would have liked to see an analysis of a complex and heterogeneous disease like CAD. Presumably, CAD cases can be linked to a combination of different risk factors, such as LDL or high blood pressure. Defining genetically defined pathways/PRS that could be correlated with the LDL and blood pressure biomarkers provided by UKB would be compelling in my view.

My overall conclusion is that the paper is interesting, showing some promises in terms of being able to address an important problem and a substantial amount of work has been done. But with that in mind, I find its presentation really challenging and the technicalities hard to follow. I would be keen to review a somewhat simplified version of this manuscript that better walks the reader through that complex evaluation process. But as of now, I struggle to provide useful insights simply because there is much that I do not understand.

Reviewer #3: Uploaded as an attachment.

**Have all data underlying the figures and results presented in the manuscript been provided?**

Reviewer #1: Yes

Reviewer #2: Yes

Reviewer #3: Yes

PLOS authors have the option to publish the peer review history of their article (what does this mean?). If published, this will include your full peer review and any attached files.

Reviewer #1: No

Reviewer #2: No

Reviewer #3: No

---

## [Decision Letter · Decision Letter 1]

18 Oct 2022

Dear Dr García-González,

Thank you very much for submitting your Methods entitled 'PRSet: pathway-based polygenic risk score analyses' to PLOS Genetics.

The manuscript was fully evaluated at the editorial level and by independent peer reviewers. The reviewers appreciated the improvements made in comparison to to your previous submission, but identified some remaining concerns that we ask you address in a revised manuscript.

We therefore ask you to modify the manuscript according to the review recommendations. Your revisions should address the specific points made by each reviewer.

Yours sincerely,

Heather J Cordell

Academic Editor

PLOS Genetics

Xiaofeng Zhu

Section Editor

PLOS Genetics

Reviewer's Responses to Questions

**Comments to the Authors:**

Reviewer #1: I find this manuscript improved from the previous version. I still have a few issues, however.

I previously raised the concern that analysis was only conducted on very large datasets, what the authors referred to as "biobank scale" data. The authors have run an additional set of anlayses on a smaller sample size saying "Our results indicate no qualitative differences in relative performance of the methods when the GWAS sample size is halved to 125k". But this is still a very large dataset and many studies are much smaller. I would suggest aplpying this to samples of 10k or 50k?

I think the issue of reliability of pathways is important here. I appreciate that comparison of the relative performance of different methods may be little impacted by this (they will likely suffer similar drops in performance). However I sitll think it's amn important point to make and that the reliability of pathways should be explcitly stated as a limitation in the conclusions.

Another reviewer asked about the overlap in SNPs between pathways. The Authors respond that this would not resolve the problem raised as since different SNPs (at the same locus) could be in different pathways and in this case such overlap would be missed. However, they could look at the correlaiotn between PRS which I think would address the issue adequately.

p.7 "Figure 2b – Source Data 1" - what is "source data 1"? It isn't mentioned in Figure 2. In fact "Source Data" are mentioned repeatedly, but I don't think this is explained anywhere.

I'm confused by Figure 2a - I don't understand how the GWAS/Target data are used. This seems particularly important given the reliance of PRSet on the size of the target sample. In the methods section for pathway enrichment the target data are not mentioned at all in relation to PRSet and only briefly for MAGMA and LDSC so it's not clear to me how these data are used differently for the different approaches. I think there needs to be a clear explanation of this, since the relative performance of the methods hinges on this.

From Fig 4b PRS and PRS-shift do not look signifcantly better than the other approaches, so this ought to be noted (though I realise that discriminatory power overall is quite low).

In the final section of the manuscript, the authors apply various PRS approaches to prediction of subgroups. I would think here that those using a single PRS for prediction (e.g. PRSice) have a disadvantage over PRSet, which applies multiple PRS (by looking at a separate PRS for each pathway). For instance if 30 pathways are considered, then PRSet is fitting 30 variables and PRSice only 1. A model built with more variables in this way will almost always provide a better fit. So it's not clear to me whether the advantage (in terms of fit measured by R^2) seen by using PRSet is due to the fitting of extra variables (multiple PRS rather than one PRS) or because, as the authors hope, the pathways themselves are informative and so improving the fit of the PRS. This could be easily investigated by randomly assigning SNPs to pathways (the same number of SNPs in each pathway, but the SNP randomly assigned) - would this give the same improvement as seen from using the 'real' pathway information? It's also not clear how many separate pathway-specific "sub-PRS" the PRS are being split into - if it's a handful of pathways it probably doesn't make much difference, but if it's 100 it may well do.

Reviewer #2: Thank you for addressing my comments and apologies for the slow review on my end. While it does remain a technical paper, I think the various edits have helped the clarity of the paper and I am happy to recommend it for publication.

Reviewer #3: The authors address comments well generally except one remaining major point on the baseline of Figure 4. See below. It is particularly appreciated that the authors made visual clues in Figure 4a, and clarified the aim of the study in Abstract and Introduction.

1.

The classification section is now a much better section. However, I do have one major concern about Figure 4. I appreciate the inclusion of PRSet-shift, while I still think the baselines of PRS included in the comparison are not the most natural way for subtypes analysis. The key step making the comparison of PRSet and other methods unfair is the subtype supervised learning step in PRSet.

To put it another way, in most disease subtype analysis (e.g. T2D: Mansour Aly et al. 2021 Nat Genee.; Depression: Peterson et al. 2018 Am J Psychiatry), the PRS for the two subtypes would be largely identical except specific regions in the genome. In this case, picking one of the PRS and using it to predict subtypes will have very poor accuracy. If I want to use PRS to predict subtypes, I will use case-case GWAS to select SNPs that are different between subtypes; if it is challenge to perform this using glmnet (in fact there are efficient alternatives such as ET-Lasso), you could constraint it to SNPs that are significant for single trait, which I believe is similar to the PRS-stratifier the authors have previously proposed. I expect similar methods would outperform PRSet. The authors suggested that the PRS-stratifier should be removed from Figure 4b as it complicates the analysis; however, I do find the PRS-stratifier is the closest to a proper benchmark for distinguishing disease subtypes. I don’t think that PRSet outperformed by methods such as the PRS-stratifier disapproves its utility. I think it is still interesting to see if PRSet reach comparable accuracy in some scenarios, as it suggest the effect sizes for SNPs within pathways are highly correlated (please note, in my previous comment “many SNPs within same pathway have correlated effect” is referring to effect size correlation, which is a connected but distinct concept than LD r2).

2.

As another comment to my major point 2, the comparison of overlapping between pathways is at gene level instead of SNP level. I don’t think it is technically challenging to map SNPs within pathways to genes and design a metric for comparison (e.g. comparing weighted sum of SNPs in cis-region). While I think there is sufficient explanation of the mechanisms in this version, I don’t think the authors are obligated to perform this analysis.

3.

Again, not obligated, but I am interested to see PRSet benched marked on MAGAMA (as it is a widely used method specifically for testing enrichment). This is related to the previous comments on MalaCard.

Line 32-33: please clarify what task does pathway PRS outperform (i.e. distinguishing subtypes).

Line 387: Where is Supplementary Note 4?

**Have all data underlying the figures and results presented in the manuscript been provided?**

Reviewer #1: **No: **

Reviewer #2: None

Reviewer #3: Yes

PLOS authors have the option to publish the peer review history of their article (what does this mean?). If published, this will include your full peer review and any attached files.

Reviewer #1: No

Reviewer #2: No

Reviewer #3: No

---

## [Decision Letter · Decision Letter 2]

16 Jan 2023

Dear Dr García-González,

Thank you very much for submitting your Research Article entitled 'PRSet: pathway-based polygenic risk score analyses' to PLOS Genetics.

The manuscript was fully evaluated at the editorial level and by independent peer reviewers. The reviewers appreciated the attention to an important topic but identified some concerns that we ask you address in a revised manuscript.

We therefore ask you to modify the manuscript according to the review recommendations. Your revisions should address the specific points made by each reviewer.

Yours sincerely,

Xiaofeng Zhu

Section Editor

PLOS Genetics

Xiaofeng Zhu

Section Editor

PLOS Genetics

Reviewer's Responses to Questions

**Comments to the Authors:**

Reviewer #1: I am happy that the authors have adequately answered my queries and recommend this for publication.

Reviewer #3: I appreciate the expansion of backgrounds in “Pathway PRSs for disease stratification”, which clarified the last major concern I raised in previous comments. The authors have improved the manuscript and specified the scope of the paper, which solved my major comments (i.e. discussion on gene overlapping analysis is more suitable for subsequent analyses). I am happy to recommend this paper for publication, with minor edits below.

Following my point on “PRS-stratifier” and after reading the response to reviewer 1’s comments “In the final section of the manuscript…”, I recommend adding 1-2 sentence to the main text section “Pathway PRSs for disease stratification”, paraphrasing following:

“We note that PRSet is a flexible model that fits multiple coefficients while the single-PRS methods only fit one coefficient. Other flexible methods could achieve similar performance (see PRS-stratifier in Supplementary Note 2). ”

Here are the reasons: I think reviewer 1’s point on “more variables for PRSet” does not question the validity of the PRSet, but rather raises the point that there are multiple approaches to improve the subtype classification when limited subtype training genotypes are available. For example, one could adjust each SNP coefficient in the “C+T” PRS to train a classification model (similar to PRS-stratifier), which is the approach adopted by analyses that use limited multi-ancestry training data for cross-ancestry PRS (e.g. PolyPred+ in Weissbrod et al. 2022 Nature Genetics). It is hard to argue what is the best approach but it is important to mention that the model flexibility itself could increase the prediction power. For example, when PRSet performs similarly to PRSet-shift, it is more likely that the model flexibility, instead of pathway information, is contributing to the improvement. It would be interesting to know if PRSet reaches similar performance as a more refined PRS-stratifier (I don’t think PRSet will outperform a method similar to PRS-stratifier, which is more flexible than PRSet), while I am happy if it is more suitable for future analyses.

**Have all data underlying the figures and results presented in the manuscript been provided?**

Reviewer #1: Yes

Reviewer #3: Yes

PLOS authors have the option to publish the peer review history of their article (what does this mean?). If published, this will include your full peer review and any attached files.

Reviewer #1: No

Reviewer #3: No

---

## [Editor Report · Decision Letter 3]

19 Jan 2023

Dear Dr García-González,

We are pleased to inform you that your manuscript entitled "PRSet: Pathway-based Polygenic Risk Score analyses and software" has been editorially accepted for publication in PLOS Genetics. Congratulations!

Yours sincerely,

Heather J Cordell

Academic Editor

PLOS Genetics

Xiaofeng Zhu

Section Editor

PLOS Genetics

Comments from the reviewers (if applicable):

**Data Deposition**

http://datadryad.org/submit?journalID=pgenetics&manu=PGENETICS-D-22-00433R3

**Press Queries**

---

## [Editor Report · Acceptance letter]

1 Feb 2023

PGENETICS-D-22-00433R3 

PRSet: Pathway-based Polygenic Risk Score analyses and software 

Dear Dr García-González, 

We are pleased to inform you that your manuscript entitled "PRSet: Pathway-based Polygenic Risk Score analyses and software" has been formally accepted for publication in PLOS Genetics! Your manuscript is now with our production department and you will be notified of the publication date in due course.

With kind regards,

Zsofi Zombor

PLOS Genetics

On behalf of:
